# TRAINING DATA SUBSET SEARCH WITH ENSEMBLE ACTIVE LEARNING

## ABSTRACT

Deep Neural Networks (DNNs) often rely on very large datasets for training. Given the large size of such datasets, it is conceivable that they contain certain samples that either do not contribute or negatively impact the DNN's optimization. Modifying the training distribution in a way that excludes such samples could provide an effective solution to both improve performance and reduce training time. In this paper, we propose to scale up ensemble Active Learning methods to perform acquisition at a large scale (10k to 500k samples at a time). We do this with ensembles of hundreds of models, obtained at a minimal computational cost by reusing intermediate training checkpoints. This allows us to automatically and efficiently perform a training data subset search for large labeled datasets. We observe that our approach obtains favorable subsets of training data, which can be used to train more accurate DNNs than training with the entire dataset. We perform an extensive experimental study of this phenomenon on three image classification benchmarks (CIFAR-10, CIFAR-100 and ImageNet), analyzing the impact of initialization schemes, acquisition functions and ensemble configurations. We demonstrate that data subsets identified with a lightweight ResNet-18 ensemble remain effective when used to train deep models like ResNet-101 and DenseNet-121. Our results provide strong empirical evidence that optimizing the training data distribution can provide significant benefits on large scale vision tasks.

## 1 INTRODUCTION

Deep Neural Networks (DNNs) have become the dominant approach for addressing supervised learning problems. They are trained using stochastic gradient methods, where (i) data is subsampled into mini-batches, and (ii) the network parameters are iteratively updated by the gradient of the parameter weights, relative to a loss function for the given labeled mini-batch of data. The architecture of the DNN, hyper-parameters of the training process, and distribution of the dataset used are all crucial ingredients that impact the final performance of the DNN.

Given the remarkable success of this recipe for supervised learning, there is significant interest in automating the end-to-end process of applying DNNs to real-world problems (Wong et al., 2018; He et al., 2018; 2019). While there has been a considerable effort towards methods and frameworks that automate DNN architecture search (Elsken et al., 2018; Luo et al., 2018; Liu et al., 2018; Xie et al., 2019; Dong & Yang, 2019) and training hyper-parameter search (Golovin et al., 2017; Mutny & Krause, 2018; Kandasamy et al., 2019); the process of searching for the right training data distribution (also called dataset curation) is still performed by experts, requiring several heuristics and significant manual effort. With the rapid growth in the availability of labeled data for large-scale vision tasks, to the order of billions of samples (Sun et al., 2017; Zeki Yalniz et al., 2019), automating the *training data subset search* would make the application of DNNs much easier for non-experts, and potentially lead to datasets and models that outperform those that were curated by hand. With the cost of data storage also becoming increasingly important, the ability to automatically identify data to remove from the training distribution is appealing from a practical perspective.

In this paper, we present a simple yet effective method to perform a training data subset search by using ensemble Active Learning (AL). The typical goal of AL is to to select, from a large unlabeled dataset, the smallest possible training set to label in order to solve a specific task (Cohn et al., 1994). We instead propose to use AL to build *data subsets* of a large labeled training dataset that give more

accurate DNNs in less training time. We tackle several key issues that have not been addressed so far by state-of-the-art AL methods. The first is the difficulty in *scaling the number of models* for ensemble AL. While it seems intuitive that more ensembles can improve performance, existing studies show no gains in AL performance beyond 10 models, and even recommend the use of only 5 models (Lakshminarayanan et al., 2017; Ovadia et al., 2019). Implicit ensemble methods for AL have been found similarly ineffective, but only evaluated in settings with with < 10 models (Beluch et al., 2018). In this study, we propose the use of implicit ensembles with hundreds of training checkpoints from different experimental runs, and empirically demonstrate the effectiveness of this approach. Second, we switch to a *large-scale experimental setting* compared to what is typically used for AL experiments. For example, Beluch et al. (2018) and Sinha et al. (2019) only acquire 40k and 64k samples at a time respectively on ImageNet, never use more than 30% of the dataset, and do not compare to the full dataset performance. In contrast, for ResNet-18 training on ImageNet, we acquire up to 500k samples at a time and use 80% of the dataset, improving top-1 accuracy by 0.5% over a model trained with the entire dataset. Third, we demonstrate the *transferability of the subsets obtained* with our approach between DNNs with very different network capacities, ranging from 10-layer ResNets (He et al., 2016) to 121-layer DenseNets (Huang et al., 2016). Given the rapid pace of model development, this shows that our data subsets remain valuable even after a particular model is surpassed by newer architectures. This has been difficult to achieve in prior AL studies (Lowell et al., 2018), and not investigated in detail for large-scale vision tasks.

To summarize, our contributions are as follows: (i) we propose a simple approach to scale up ensemble AL methods to hundreds of models with a negligible computational overhead at train time; (ii) we evaluate several methods to reduce the size of existing large datasets with AL; and (iii) we conduct a detailed empirical study on three popular image classification benchmarks, studying the impact of key design choices, and the robustness of the selected subsets to changes in model architecture. Our study has significant practical implications for DNN training with stochastic gradient methods, as well as for storage of large training datasets.

## 2 BUILDING DATA SUBSETS WITH ACTIVE LEARNING

Consider a distribution $p(\mathbf{x}, y)$ over inputs $\mathbf{x}$ and labels $y$. In a Bayesian framework, the *predictive uncertainty* of a particular input $\mathbf{x}^*$ after training on a dataset $L$ is denoted as $P(y = k|\mathbf{x}^*, L)$. The predictive uncertainty will result from *data (aleatoric) uncertainty* and *model (epistemic) uncertainty* (Kendall & Gal, 2017). A model's estimates of *data uncertainty* are described by the posterior distribution over class labels $y$ given a set of model parameters $\theta$. This is typically the softmax output in a classification DNN. Additionally, the *model uncertainty* is described by the posterior distribution over the parameters $\theta$ given the training data $L$ (Malinin & Gales, 2018):

$$\underbrace{P(y = k|\mathbf{x}^*, L)}_{Predictive} = \int \underbrace{P(y = k|\mathbf{x}^*, \theta)}_{Data} \underbrace{p(\theta|L)}_{Model} \, d\theta. \tag{1}$$

We see that, uncertainty in the model parameters $p(\theta|L)$ induces a distribution over the softmax distributions $P(y = k|\mathbf{x}^*, \theta)$. The expectation is obtained by marginalizing out the parameters $\theta$. Unfortunately, obtaining the full posterior $p(\theta|L)$ using Bayes' rule is intractable. If we train a single DNN, we only obtain a single sample from the distribution $p(\theta|L)$. Ensemble uncertainty estimation techniques approximate the integral from Eq. 1 by Monte Carlo estimation, generating multiple samples using different members of an ensemble (Lakshminarayanan et al., 2017):

$$P(y = k|\mathbf{x}^*, L) \approx \frac{1}{E} \sum_{e \in E} P(y = k|\mathbf{x}^*, \theta_e), \ \theta_e \sim q(\theta). \tag{2}$$

where $q(\theta)$ represents an approach used for building the ensemble. The strength of the ensemble approximation depends on both the number of samples drawn ($E$), and how the parameters for each model in the ensemble are sampled (i.e., how closely the distribution $q(\theta)$ matches $p(\theta|L)$).

---

**Algorithm 1** Building Data Subsets with Active Learning.

---

Initialize parameters of acquisition model $\{\theta_a^{(e)}\}_{e=1}^E$ and subset model $\theta_s$     ▷ **initialization**
Initialize data subset $S$
Compute acquisition function $\alpha(\mathbf{x}, \mathcal{M}_a) \forall \mathbf{x} \in L$     ▷ **acquisition**
Append $N_s$ samples with the maximum acquisition function to $S$
Use $S$ to optimize parameters $\theta_s$ of subset model     ▷ **optimization**

---

Uncertainty estimation acts as the core component of our approach to build data subsets, for which the pseudo code is presented in Algorithm 1. The algorithm is a modification of the typical AL setting, and involves the following:

1. A **labeled dataset**, consisting of $N_l$ labeled pairs, $L = \{(\mathbf{x}_l^j, y_l^j)\}_{j=1}^{N_l}$, where each $\mathbf{x}^j \in X$ is a data point and each $y^j \in Y$ is its corresponding label.

2. An **acquisition model**, $\mathcal{M}_a : X \to Y$. For our ensemble AL approach, the acquisition model $\mathcal{M}_a$ takes the form of a set of $E$ different DNNs with parameters $\{\theta_a^{(e)}\}_{e=1}^E$.

3. A **data subset**, $S = \{(\mathbf{x}_s^j, y_s^j)\}_{j=1}^{N_s}$, where $S$ is a subset of $L$ selected using an acquisition function $\alpha(\mathbf{x}, \mathcal{M}_a)$.

4. A **subset model**, $\mathcal{M}_s : X \to Y$, with parameters $\theta_s$, trained on $S$.

As detailed in Algorithm 1, we decouple the data selection and final optimization of AL into two different models. The *acquisition model* selects a subset of data using ensemble uncertainty estimation, which is then used to optimize the parameters of the *subset model*.

## 2.1 INITIALIZATION SCHEMES

We consider three different *initialization schemes* for the parameters of the acquisition and subset models: pretrain, compress and build up. The pretrain scheme uses the entire dataset $L$ for pretraining both the acquisition and subset models. During optimization, the subset model is then finetuned on the data subset $S$. In the compress scheme, the acquisition model is pretrained on $L$ but the subset model is randomly initialized and trained from scratch on $S$. The acquisition model therefore accesses all the data and then 'compresses' the dataset for the subset model. For the pretrain and compress schemes, the subset $S$ is initialized with an empty set.

Finally, in the build up scheme, we follow an iterative AL loop. Specifically, we start by initializing $S$ with a randomly selected subset of the data to train an acquisition model. After performing acquisition, instead of training a single subset model, we optimize an ensemble of $E$ networks. This ensemble is used as an acquisition model for a subsequent iteration. Our goal is to finally reach a subset of $N_s$ samples. As observed by Chitta et al. (2018a), exponentially growing the dataset size offers practical benefits in an AL loop setting. We therefore follow this approach, by initializing $S$ with $\frac{N_s}{8}$ random samples, and iterating two further times at $\frac{N_s}{4}$ and $\frac{N_s}{2}$ samples before obtaining a final subset of size $N_s$.

## 2.2 ACQUISITION FUNCTIONS

In our experiments, we empirically evaluate four acquisition functions of the form $\alpha : \mathbf{x} \to \mathbb{R}$ for AL– entropy, mutual information, variation ratios and error count. We choose these well-known acquisition functions to maintain simplicity and scalability. For a detailed theoretical analysis of these acquisition functions, we refer the reader to Gal (2016).

Bayesian AL approaches argue that the acquisition function must target samples with high *model uncertainty* rather than *data uncertainty*. Intuitively, the reasoning behind this is that *model uncertainty* is a result of not knowing the correct parameters, which can be reduced by adding the right data and retraining the model. However, *data uncertainty* exists even for a model with the most optimal parameters, and cannot be reduced by adding more data (Kendall & Gal, 2017).

In the case of classification, the predictive uncertainty for a sample $P(y = k|\mathbf{x}^*, L)$ from Eq. 1 is a multinomial distribution, which can be represented as a vector of probabilities $\mathbf{p}$ over each of the $K$ classes. We can obtain the predictive uncertainty for a sample as its entropy (Shannon, 1948):

$$\mathcal{H}(\mathbf{p}) = -\mathbf{p}^T \log \mathbf{p}. \tag{3}$$

However, once we marginalize out $\theta$ from Eq. 1 it is impossible to tell if this *predictive uncertainty* is a result of *model uncertainty* or *data uncertainty*. One acquisition function that explicitly looks for large disagreement between the models (i.e., *model uncertainty*), is mutual information (Houlsby et al., 2011; Smith & Gal, 2018), also called BALD and Jensen-Shannon Divergence:

$$\underbrace{\mathcal{J}(\mathbf{p})}_{Model} = \underbrace{\mathcal{H}(\mathbf{p})}_{Predictive} - \underbrace{\frac{1}{E} \sum_{e \in E} \mathcal{H}(\mathbf{p}^{(e)})}_{Data}. \tag{4}$$

where $\mathbf{p}^{(e)}$ denotes the prediction of an individual member of the ensemble before marginalization. Since entropy is always positive, the maximum possible value for $\mathcal{J}(\mathbf{p})$ is $\mathcal{H}(\mathbf{p})$. However, when the models make similar predictions, $\frac{1}{E} \sum_{e=1}^{E} \mathcal{H}(\mathbf{p}_e) \to \mathcal{H}(\mathbf{p})$, and $\mathcal{J}(\mathbf{p}) \to 0$, which is its minimum value. This shows that $\mathcal{J}$ encourages samples with high disagreement to be selected during the data acquisition process. An alternate way to look at the metric is that from the predictive uncertainty, we subtract away the expected *data uncertainty*, leaving an approximate of the *model uncertainty* (Depeweg et al., 2017).

Variation ratios (Gal, 2016) is another acquisition function that looks for disagreement. It is defined as the fraction of members in the ensemble that do not agree with the majority vote $M = \underset{e \in E}{\text{Mode}}(\underset{k \in K}{\arg\max}\, \mathbf{p}_k^{(e)})$:

$$\mathcal{V}(\mathbf{p}) = 1 - \frac{1}{E} \sum_{e \in E} (\underset{k \in K}{\arg\max}\, \mathbf{p}_k^{(e)} = M), \tag{5}$$

where $K$ is the number of classes. This is the simplest quantitative measure of variation, and prior applications in literature show that it works well in practice (Gal et al., 2017; Beluch et al., 2018; Chitta et al., 2018b). Our work focuses on classification due to its prevalence in the large-scale setting, but we note that $\mathcal{V}$ shares similarities to variance and standard deviation, which are used as acquisition functions in the Bayesian setting for regression (Tsymbalov et al., 2018).

Our final acquisition function, error count, is similar to variation ratios, but checks for disagreement with the ground truth label $y$ rather than the mode of predictions $M$:

$$\mathcal{E}(\mathbf{p}) = 1 - \frac{1}{E} \sum_{e \in E} (\underset{k \in K}{\arg\max}\, \mathbf{p}_k^{(e)} = y). \tag{6}$$

This is typically not used in AL experiments as it cannot be computed without the ground truth labels. In our setting, where the labels are available, we use this function as a baseline which, in principle, prioritizes mistakes made by the network when selecting data.

## 2.3 Ensemble Configurations

State-of-the-art ensemble-based AL approaches use different random seeds to construct ensembles (Beluch et al., 2018). They recommend the number of samples drawn to be in the range $E \in (5, 10)$ models (Lakshminarayanan et al., 2017). In theory, the error of a Monte Carlo estimator should decrease with more samples, which is evident in other BNN based uncertainty estimation techniques, that require the number of stochastic samples drawn to be increased to the range $E \in (50, 100)$ (Gal & Ghahramani, 2015).

The major limiting factor preventing the training of $E \in (50, 100)$ models with different random seeds for ensemble AL is the computational burden at train time. Implicit ensembling approaches that are computationally inexpensive, such as Dropout (Srivastava et al., 2014), suffer from mode

collapse, where the different members in the ensemble lack sufficient diversity for reliable uncertainty estimation (Pop & Fulop, 2018). An alternate approach, called snapshot ensembles, that is less computationally expensive at train time, uses a cyclical learning rate to converge to multiple local optima in a single training run (Huang et al., 2017). However, this technique is also limited to ensembles in the range of $E = 6$ members. In our work, we present an implicit ensembling approach that allows users to draw a large number of samples using the catastrophic forgetting property in DNNs (Toneva et al., 2018). Specifically, we exploit the disagreement between different checkpoints stored during successive training epochs to efficiently construct large and diverse ensembles. We collect several training checkpoints over multiple training runs with different random seeds.

Though we study the impact of the number of samples, our work does not attempt specifically to better match q($\theta$) from Eq. 1 and p($\theta|L$) from Eq. 2, which is an open and active research area (Malinin & Gales, 2018). The overall algorithm is generic and can potentially benefit from more advanced techniques for uncertainty estimation.

## 3 EXPERIMENTS

In this section, we demonstrate the effectiveness of AL for building data subsets of three image classification benchmarks. We initially investigate the impact of the initialization schemes discussed in Section 2.1 and acquisition functions from Section 2.2 for the ResNet-18 architecture (He et al., 2016). We then focus on scaling up the ensemble and evaluating the robustness of our subsets to architecture shifts. We experiment with three datasets: CIFAR-10 and CIFAR-100 (Krizhevsky, 2009), as well as ImageNet (Deng et al., 2009). The CIFAR datasets involve object classification tasks over natural images: CIFAR-10 is coarse-grained over 10 classes, and CIFAR-100 is fine-grained over 100 classes. For both tasks, there are 50k training images and 10k validation images of resolution $32 \times 32$, which are balanced in terms of the number of training samples per class. ImageNet consists of 1000 object classes, with annotation available for 1.28 million training images and 50k validation images of resolution $256 \times 256$. This dataset has a slight class imbalance, with 732 to 1300 training images per class.

### 3.1 IMPLEMENTATION DETAILS

Unless otherwise specified, we use 8 models with the ResNet-18 (He et al., 2016) architecture to build the acquisition and subset models. For ImageNet, each ResNet-18 uses the standard kernel sizes and counts. For CIFAR-10 and CIFAR-100, we use a variant of ResNet-18 as proposed in He et al. (2016). For all three tasks, we do mean-std pre-processing, and augment the labeled dataset on-line with random crops and horizontal flips. Optimization is done using Stochastic Gradient Descent with a learning rate of 0.1 and momentum of 0.9, and weight decay of $10^{-4}$. On CIFAR, we use a patience parameter (set to 25) for counting the number of epochs with no improvement in validation accuracy, in which case the learning rate is dropped by a factor of 0.1. We end training when dropping the learning rate also gives no improvement in the validation accuracy after a number of epochs equal to twice the patience parameter. If the early stopping criterion is not met, we train for a maximum of 400 epochs. On ImageNet, we train for a total of 150 epochs, scaling the learning rate by a factor of 0.1 after 70 and 130 epochs. Experiments are run on Tesla V100 GPUs.

### 3.2 RESULTS

**Initialization schemes.** In our first experiment, we compare the three initialization schemes introduced in Section 2.1 to a random subsampling baseline. For this experiment, we fix the number of ensemble members to 8 for CIFAR and 4 for ImageNet, each with a different random seed. We fix the acquisition function to Mutual Information ($\mathcal{J}$), as defined in Eq. 4. For the pretrain scheme, we pretrain 8 (or 4) models with different random seeds on the entire dataset $L$, and finetune them starting with a learning rate of $10^{-3}$ on the chosen subset $S$. For the other schemes, we train the subset model from scratch on $S$. We report the top-1 validation accuracy of three independent ensembles, each from a different experimental trial, plotting the mean with one standard deviation as an error bar. These results are summarized in Fig. 1.

We observe certain common trends for all three datasets: random subsampling (blue in Fig. 1) leads to a steady drop-off in performance; and the pretrain scheme (orange in Fig. 1) does not significantly

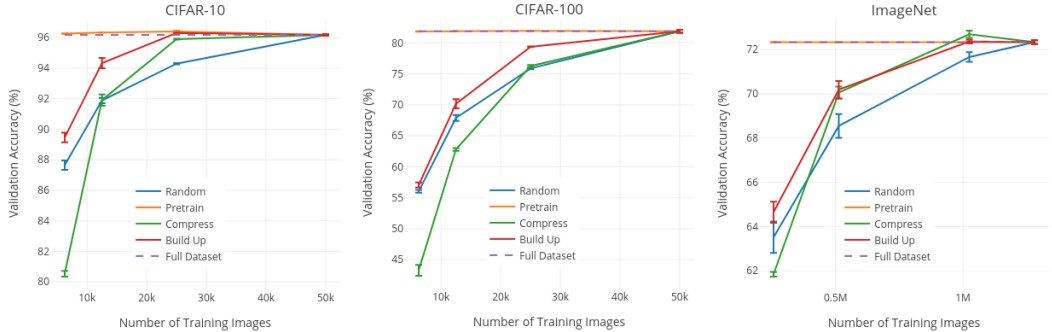

Figure 1: Comparison of pretrain, compress and build up initialization schemes on CIFAR-10, CIFAR-100 and ImageNet. Results shown are mean and std over three trials of the top-1 validation accuracy of an ensemble of ResNet-18 models. While compress does not perform well on small subsets of data, build up provides strong results on all subset sizes for all three tasks.

impact the performance in comparison to training with the full dataset. Interestingly, the compress scheme (green in Fig. 1) performs extremely poorly when the subset chosen is very small, but performs well when the acquisition and subset models have a similar overall dataset size (eg., 1M samples on ImageNet). The poor performance of the compress scheme implies that the uncertainty estimates for this experiment are not robust to large changes in the dataset distribution between the acquisition and subset models. The build up scheme (red in Fig. 1) consistently outperforms random subsampling by a large margin. The performance is robust across all three tasks. Based on these observations, we fix the initialization scheme to build up for the next experiments. With a sufficiently large subset of the data (eg., 25k on CIFAR-10, 1M on ImageNet), the build up scheme slightly outperforms a model trained on the full dataset.

**Acquisition functions.** In our next experiment, we compare random subsampling against the four acquisition functions from Section 2.2. Similar to the previous experiment, we use ensembles of 8 members for CIFAR and 4 members for ImageNet for both the acquisition and subset models. We run three experimental trials, and report the mean validation accuracy of the subset model ensemble for each trial, in Table 1.

For all four acquisition functions, the subset models significantly outperform the baseline (random) at the final iteration. Further, when using 50% of the data on CIFAR-10, and 80% of the data on CIFAR-100 and ImageNet, we obtain subsets of data that improve performance compared to training on the full dataset (100%). Among the four functions, mutual information ($\mathcal{J}$) and variation ratios ($\mathcal{V}$) outperform entropy ($\mathcal{H}$) and error count ($\mathcal{E}$). Interestingly, though the $\mathcal{E}$ acquisition function is similar to $\mathcal{V}$, and also has access to the ground truth labels, the data selected by it in the first iteration leads to poor performance, which is not completely recovered in the subsequent iterations. This indicates that a sample for which all the ensemble members collectively make an error may be 'too difficult' and therefore not an ideal choice for the training dataset.

Among the two best acquisition functions, on the CIFAR-10 dataset, $\mathcal{J}$ outperforms $\mathcal{V}$. This is because there are very few absolute disagreements in this setting due to the small number of classes and high performance, which leads to a very small number of samples with non-zero values for $\mathcal{V}$. However, on the CIFAR-100 and ImageNet datasets, with more classes and lower overall performance, there is a greater number of disagreements, and $\mathcal{V}$ outperforms $\mathcal{J}$ by a significant amount when using 80% of the dataset. Compared to training on the full dataset, we obtain a 0.5% absolute improvement in validation accuracy on both CIFAR-100 and ImageNet when using the subset acquired by $\mathcal{V}$. Additionally, this performance improvement is accompanied by a 20% reduction in training time on both these tasks.

**Ensemble configurations.** We now explore the possibility of further gains in performance by scaling up the ensemble to increase the number of samples drawn in the Monte Carlo estimator as per Eq. 2. For the remaining experiments, we focus on the final AL iteration for the ImageNet dataset as per the build up scheme. To do this, we start by setting up 5 different training runs on 40% of the

Table 1: Top-1 validation accuracy (in %) of ensembles trained on subsets of CIFAR-10, CIFAR-100 and ImageNet (mean of 3 trials), comparing different acquisition functions. Each column indicates a subset with a size equal to the specified percentage of the full dataset. Mutual information ($\mathcal{J}$) and variation ratios ($\mathcal{V}$) give the best performance.

| Dataset | CIFAR-10 | | | CIFAR-100 | | | ImageNet | | |
|---|---|---|---|---|---|---|---|---|---|
| Acquisition Function | 12.5% | 25% | 50% | 20% | 40% | 80% | 20% | 40% | 80% |
| Random | 87.65 | 91.88 | 94.30 | 63.96 | 74.18 | 80.65 | 63.51 | 68.54 | 71.66 |
| Entropy ($\mathcal{H}$) | 89.67 | 94.29 | 96.08 | 65.83 | 76.35 | 81.94 | 64.11 | 69.64 | 72.00 |
| Mutual Information ($\mathcal{J}$) | 89.46 | 94.34 | **96.30** | 66.11 | 76.29 | 82.02 | 64.65 | 70.18 | 72.36 |
| Variation Ratios ($\mathcal{V}$) | 89.86 | 94.42 | 95.76 | 65.28 | 76.27 | **82.37** | 64.39 | 69.20 | **72.78** |
| Error Count ($\mathcal{E}$) | 87.30 | 94.02 | 96.08 | 65.37 | 76.13 | 82.06 | 58.13 | 64.66 | 72.10 |
| Full Dataset (**100%**) | 96.18 | | | 81.86 | | | 72.33 | | |

Table 2: Effect of increasing the number of models in an acquisition ensemble trained with 40% of ImageNet. Results shown are top-1 accuracy evaluated on the 40% selected data (512k samples used for training) and 60% unselected data (768k remaining samples). Gains in performance on the unselected data show that large scale checkpoint-based ensembles exhibit high diversity.

| Eval Set | Single (1) | Seeds (5) | Checkpoints (5) | Checkpoints (20) | Combined (100) |
|---|---|---|---|---|---|
| Selected | 58.10 | 74.98 | 72.59 | 79.60 | 83.78 |
| Unselected | 70.85 | 82.55 | 81.47 | 84.02 | 85.57 |

ImageNet dataset (512k samples) as selected by the best performing acquisition function in Table 1 ($\mathcal{J}$). For each of these 5 training runs, we store the 21 checkpoints obtained in the final stage of training (epochs 130-150, exact implementation details provided in appendix). We pick 4 ensemble configurations from these ResNet-18 training runs to utilize as the acquisition model for an ablation study: (i) random seeds, which uses a total of 5 models from the best performing epoch of each run; (ii) 5 checkpoints, which uses the 5 models from epochs 130:5:150 of a single run; (iii) 20 checkpoints, which uses the 20 models from epochs 131:1:150 of a single run; and (iv) combined, which uses the '20 checkpoints' combined over all 5 runs to give 100 models.

We initially evaluate the performance of these four ensemble configurations on the data available for sampling. To this end, we report the top-1 accuracy of the four ensemble configurations, along with a baseline of a single model, when evaluated on the 40% selected data (i.e, the training set) and 60% unselected data of ImageNet. Our results are shown in Table 2. Results for the Single (1), Checkpoints (5) and Checkpoints (20) columns use the best seed of 5 runs. As the number of members in the ensemble is scaled up, we observe large and clear improvements in performance on both selected and unselected data. This shows that the checkpoints obtained with no additional computational cost at train time can be used to generate diverse ensembles. A detailed analysis of the diversity and agreement between checkpoints is provided in the appendix.

The significant gains (around 3%) in performance on unselected data in Table 2 as the number of models is increased from 5 to 100 indicates the potential for better sampling by scaling up. It is additionally worth noting that for all ensemble configurations, the performance is better on the larger unselected subset of data than on the selected training set. For a single model, the gap in top-1 accuracy between these two subsets is nearly 13%. This demonstrates the huge amount of redundancy in the unselected part of the dataset as a result of our AL based selection.

Further, we are interested in how the ensemble configurations affect the acquisition function. To this end, we query for an additional 40% of the unselected data (512k samples) using the variation ratios ($\mathcal{V}$) acquisition function, with each of the four ensemble configurations. We obtain four new subsets, each with 80% of the samples in ImageNet. The top-1 validation accuracy of a subset model trained using each of these new subsets is shown in Table 3, along with a baseline of random sampling of 80% of the data. We observe a steady increase in performance as the number of models in $\mathcal{M}_a$ during acquisition is increased, showing the benefits of scaling up ensemble AL. In particular, the combined configuration improves top-1 accuracy over random sampling by 1.1%. Note that the earlier results in Table 1 use an ensemble of 4 models for evaluation; but Table 3 always evaluates

Table 3: Effect on the subset model of increasing the number of ensemble members in the acquisition model using the $\mathcal{V}$ acquisition function. Results shown are top-1 validation accuracy of a single ResNet-18 model trained using 80% of the ImageNet dataset selected with different ensemble configurations, compared to a baseline of random acquisition. Best results are obtained by the combined configuration, which uses different random seeds and training checkpoints in the acquisition model.

| Random | Seeds (5) | Checkpoints (5) | Checkpoints (20) | Combined (100) |
|--------|-----------|-----------------|------------------|----------------|
| 69.24  | 69.97     | 70.10           | 70.18            | 70.34          |

Table 4: Transferring the 80% subset of ImageNet using ResNet-10 and ResNet-18 to new architectures for the subset model. We observe that these subsets provide near equivalent performance to training on the entire dataset (Full-100) across all new architectures.

| Dataset   | ResNet-18 | ResNet-34 | ResNet-50 | ResNet-101 | DenseNet-121 |
|-----------|-----------|-----------|-----------|------------|--------------|
| Random-80 | 69.24     | 73.00     | 75.15     | 76.72      | 74.59        |
| AL-R10-80 | 70.31     | 73.49     | 75.91     | 77.87      | 75.28        |
| AL-R18-80 | 70.34     | 73.61     | 76.18     | 77.75      | 75.42        |
| Full-100  | 70.12     | 73.68     | 76.30     | 77.99      | 75.30        |

a single model, to allow for a fair comparison between the datasets acquired with 5 models vs. 100 models. Our results show that scaling up is key to exploiting implicit ensembling techniques for AL. Existing work that uses the related idea of snapshot ensembles for AL performs poorly (Beluch et al., 2018). This is likely due to (i) the small number of models used, and (ii) the smaller-scale experimental setting (only 2k samples acquired each iteration).

**Robustness to architecture shift.** Finally, we evaluate the robustness of the subsets to changes in model capacity. Specifically, we evaluate the robustness of the best performing subset selected with a ResNet-18 acquisition model in Table 3 ('combined'). Additionally, we consider the subset obtained with an even more lightweight ResNet-10 model, where we remove 2 convolutional layers from each residual block of a ResNet-18. We use these subsets (referred to as AL-R18-80 and AL-R10-80) to train subset models with the ResNet-18, ResNet-34, ResNet-50, ResNet-101 (He et al., 2016) and DenseNet-121 (Huang et al., 2016) architectures. For reference, we also evaluate a randomly subsampled 80% of ImageNet (Random-80) and the full dataset (Full-100). Our results are summarized in Table 4. As shown, on all 5 architectures, the subset obtained by AL with ResNet-18 achieves similar performance to training on the full dataset, with a 20% reduction in overall training time. The AL-R10-80 subset also strongly outperforms the random baseline. This ability to transfer selected subsets to larger architectures has significant implications in domains where training time is crucial, such as MLPerf (MLP, 2019).

## 4 RELATED WORK

A comprehensive review of classical approaches to AL is presented in Settles (2010). In these approaches, data samples for which the current model is uncertain are queried for labeling. It is occasionally observed in the classical AL literature that training on a subset of data can give better models than training on the full dataset (Schohn & Cohn, 2000; Lapedriza et al., 2013; Gavves et al., 2015; Konyushkova et al., 2015). However, this is not investigated with state-of-the-art AL techniques for DNNs (Beluch et al., 2018; Sinha et al., 2019). These methods typically assume that the performance obtained by training on the entire data pool is an upper bound. We show that the full dataset is not an upper bound, demonstrating distinct advantages of training on data subsets when they are sufficiently large.

Despite the widespread use of image classification datasets, there has only recently been an increased interest in understanding the properties of data subsets. Core-set selection (Sener & Savarese, 2017) is one such attempt to reduce dataset sizes, by finding a representative subset of points based on relative distances in the DNN feature space. Vodrahalli et al. (2018) also aims to find representative subsets, using the magnitude of the gradient generated by each sample for the DNN as an impor-

tance measure. It is important to note that these techniques are *unable to match or improve* the performance of a model trained with all the data for DNNs.

More recent techniques are able to successfully reduce dataset sizes by identifying redundant examples, albeit to a much smaller extent than our approach. Birodkar et al. (2019) uses clustering in the DNN feature space to identify redundant samples, leading to a discovery of 10% redundancy in the CIFAR-10 and ImageNet datasets. Select via proxy (Coleman et al., 2019) uses simple uncertainty metrics (confidence, margin and entropy) to select a subset of data for training, and removes 40% of the samples in CIFAR-10 without reducing the network performance. In comparison to these methods, our technique not only maintains, but also improves the performance of a DNN. Further, we empirically show that this can be done with much less data than these methods, for example, on CIFAR-10, we only require 50% of the training dataset.

A second category of works on data subsets is closely related to catastrophic forgetting in DNNs. Toneva et al. (2018) uses the number of instances in which a previously correctly classified sample is 'forgotten' and misclassified during training as an importance measure. By doing so, this method is able to remove 30% of the samples that are 'unforgettable' from CIFAR-10 without significantly reducing performance. Chang et al. (2017) propose a similar idea of emphasizing data points whose predictions have changed most over the previous epochs during training. Rather than directly subsampling, this variance in predictions is used to increase or decrease the sampling weight during training, and therefore the approach has no significant impact on training time. In comparison, by training on only a specific subset, we not only improve performance but also cut down training time by 20% to 50% for the datasets used in our study.

## 5 CONCLUSION

In this paper, we presented an approach to build data subsets for deep neural networks. Our method uses ensemble Active Learning to estimate the uncertainty of each sample in a dataset, and then chooses only the highest uncertainty samples for training. Our results demonstrate that this improves the performance of a DNN compared to training with the entire dataset on three different image classification benchmarks. Moreover, we propose a simple technique to scale up ensembles leading to additional accuracy gains with minimum computational overhead. Importantly, our results demonstrate that datasets obtained using AL can be effectively reused for training new models with different capacity or network architectures.

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

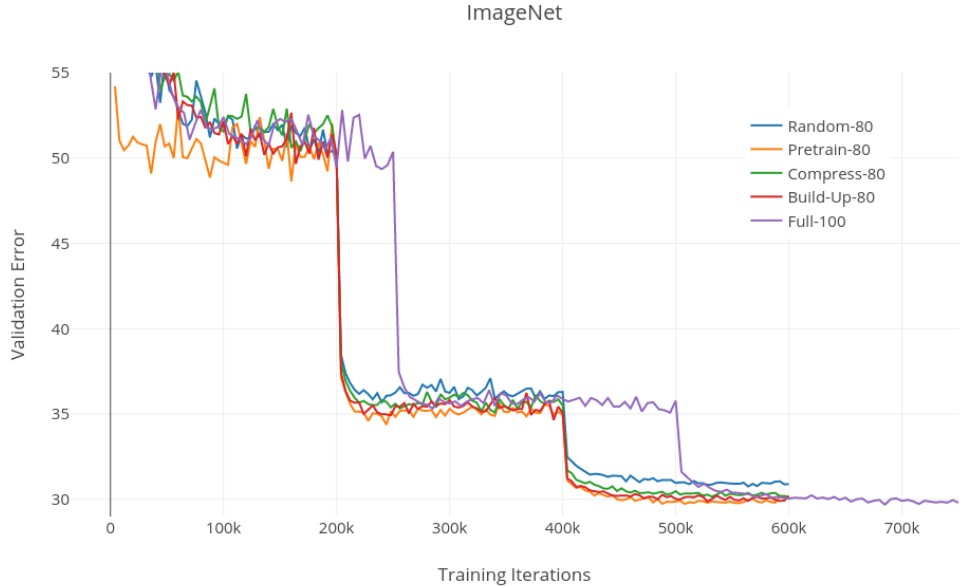

Figure 2: Comparing initialization schemes for ImageNet ResNet-18 training with variation ratios. All three initialization achieve similar performance. Compared to training on the full dataset, training time is reduced by 20%.

Table 5: Effect of leaving out the highest uncertainty samples as outliers while sampling 50% of the CIFAR-10 and CIFAR-100 datasets with the compress initialization scheme and mutual information acquisition function. Results shown are the mean top-1 validation accuracy over 3 trials. We observe that the highest uncertainty samples are outliers on CIFAR-100, but this is not the case on CIFAR-10.

| CIFAR-10 | | | CIFAR-100 | | |
|---|---|---|---|---|---|
| No Outliers | 12.5% Outliers | 25% Outliers | No Outliers | 12.5% Outliers | 25% Outliers |
| **95.77** | 94.85 | 93.81 | 75.89 | **76.76** | 76.41 |

## A  ADDITIONAL EXPERIMENTS

**Validation curves for different initialization schemes.** To analyze the runtime and convergence of different models, we plot the top-1 validation error of a single ResNet-18 model for the three initialization schemes from Section 2.1. This experiment is run using the variation ratios acquisition function on 80% of ImageNet in three settings: Pretrain-80, Compress-80 and Build-Up-80. We compare to the baselines of 80% randomly sampled data (Random-80) and the full dataset (Full-100). Our results are shown in Fig. 2. Initially, the error for Pretrain-80 is lower than the other approaches, but towards the end of training, Pretrain-80, Compress-80 and Build-Up-80 obtain a similar validation accuracy. All three initialization schemes finish training in 20% less time than using the full dataset.

**Outliers.** It is possible that a very small subset with high uncertainty is informative to the acquisition model, but *too difficult* for the subset model which is trained from scratch with just these samples. To check for this, we repeat the experiment for the compress scheme on CIFAR from Fig. 1 while choosing a subset of 25k samples (50%), but set aside a percentage of the highest uncertainty samples as outliers instead of adding them to $S$ them during acquisition. For example, for 12.5% outliers, after sorting by the acquisition function, we select the samples in the range 37.5% to 87.5% as the subset $S$ instead of 50% to 100%. As shown in Table 5, leaving outliers does not improve the performance of the compress scheme on CIFAR-10. Even in the case of CIFAR-100, where there is an improvement in performance after removing outliers, the obtained accuracy of 76.76% is well

Table 6: Effect of class-balanced random sampling on the ImageNet dataset. Results shown are the top-1 validation accuracy of a single trial for training a single ResNet-18 model. We observe that class balancing does not have significant impact on performance.

| Random | | | Random with Class Balance | | |
|---|---|---|---|---|---|
| **10%** | **20%** | **40%** | **10%** | **20%** | **40%** |
| 49.57 | 57.76 | 64.43 | 49.10 | 57.80 | 64.34 |

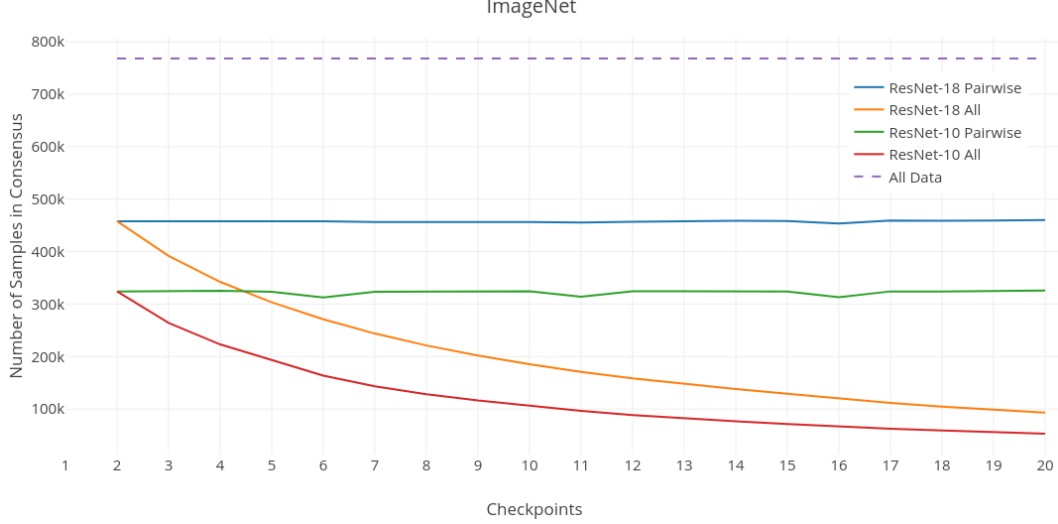

Figure 3: Analyzing the prediction consensus between different checkpoints during ImageNet training. We observe very little, but consistent agreement in predictions between any consecutive pair of checkpoints. The overall agreement among the group of models decays rapidly. With 20 checkpoints, there is only consensus on around 90k samples for ResNet-18 and 50k samples for ResNet-10 out of the 768k samples used for this evaluation.

short of the build up scheme (red in Fig. 1), which reaches 79.37%. This indicates that though the highest uncertainty samples are sub-optimal for training when taken alone, they are important when used as part of a larger set of samples.

Additionally, we conduct an experiment checking for outliers on ImageNet with the build up scheme in the setting of Table 3. In this experiment, we sample 80% of the dataset while leaving out 50k samples (approximately 4%) as outliers for the 'combined' ensemble configuration. Doing so reduces subset model performance from 70.34% to 70.09%, indicating that the highest uncertainty samples are indeed crucial to outperforming the model trained on the full dataset.

**Class-balanced sampling on ImageNet.** While CIFAR-10 and CIFAR-100 have a balanced number of samples per class, ImageNet has a slight class imbalance. This allows a potentially more refined baseline than pure random sampling, by additionally incorporating class balance. We implement this approach by collecting 128, 256 and 512 samples per class randomly for all 1000 classes. This leads to datasets of 10%, 20% and 40% of the full dataset. We compare this baseline to standard random sampling in terms of top-1 validation accuracy for ResNet-18 training. Our results are summarized in Table 6. We observe no significant differences between standard random sampling and the new random sampling with class balance.

**Checkpoint consensus.** We extend our analysis on the 768k unselected samples of ImageNet from the setting in Table 2, by checking the consensus between the group of $n$ final training checkpoints using both the ResNet-18 and ResNet-10 architectures for $n = 2$ to $20$. In addition, we check the consensus in predictions for every consecutive pair of checkpoints as a reference. Our results are presented in Fig. 3. We observe that for ResNet-18 training, any consecutive pair of checkpoints

only agree on the top-1 prediction of around 450k samples (blue in Fig. 3, which is 58% of the data). All 20 checkpoints of a single run only agree on 90k samples (orange in Fig. 3, which is 11.7% of the data). This is surprising, since the top-1 accuracy of each model on the unselected data is above 70%, indicating that though they are all from the same training run and have similar accuracy, each checkpoint makes different kinds of errors. These results also provide further support to the findings of Toneva et al. (2018), which show that a very small subset of samples are 'unforgettable' once learned by the network, and many samples are repeatedly relearned and forgotten. Since the number of disagreements provides valuable information about the uncertainty of each sample for the acquisition functions used in our study, these results showcase the benefits of using more models in an ensemble for uncertainty estimation. The consensus trends for both pairwise and all models remain similar for the ResNet-10 architecture, though the absolute values are lower than ResNet-18 due to the lower model accuracy (green and red in Fig. 3).

