# OpenReview forum: "Training Data Distribution Search with Ensemble Active Learning"
_ICLR.cc/2020/Conference — Reject_

### Official Review · AnonReviewer3 · 2019-10-22
**Official Blind Review #3**

**Rating:** 6

**Review:**

This work makes use of uncertainty estimation methods from active learning to select a subset of training data that produces models with similar (or better) performance compared to models trained on the full training set. It proposes a way to improve the Monte Carlo estimation of model uncertainty by including multiple checkpoints that are generated "for free" during a training run, thereby increasing the number of samples from 5-10 in previous work to 100 in this work. It compares several initialization schemes for the subset model using mutual information as the acquisition function, finds that a "build-up" approach (based on Chitta et. al 2018a) works best, and uses that for the rest of the studies. It then compares several acquisition functions, using the build-up approach, finds that variation ratio performs best, and uses that for the rest of the studies. Next, it compares the Top-1 accuracy on ImageNet obtained by evaluating the ensemble models produced by different ensembling schemes, and finds that ensembling 20 checkpoints from 5 training runs with different random seeds work best. Then, it uses acquisition models that use ensembles from each ensembling scheme to select subsets of the ImageNet data to be used for training the subset model, and then compares the performance of the subset models. Finally, it demonstrates this method of selecting a subset of the training data works even if the subset is used to train a model with a different architecture from the acquisition model.

Strengths:
- Algorithm is likely to be useful in practice. Training dataset can be "compressed" using a smaller architecture like ResNet-18, then used to train larger architectures like DenseNet-121, thus saving the amount of compute per training epoch. Using training checkpoints in the ensemble is very practical but not obvious (to me), since my first intuition would be that checkpoints from the same training run would not provide enough diversity to improve the acquisition function. I am glad that there were thorough experiments to address this concern and demonstrate that it works.
- Experiments answer key questions about the method proposed, and the sequence of experiments have a clear logical flow. Good baselines. Clear notation and problem set-up.

Weakness that affected the score:
- Missing detail on the build-up initialization scheme. The work referred to Chitta et al., 2018a, but that algorithm requires selecting a growth parameter. This growth parameter determines the number of times the subset model needs to be retrained, which can affect the viability of this method in practice. I would like to see the build-up initialization scheme described in greater detail.

Clarifications:
- In Table 2 and Table 3, are the results in the "Single (1)",  "Checkpoints (5)", and "Checkpoints (20)" columns obtained by averaging over the 5 random seeds?
- In the last column of Table 2, Top-1 accuracy of ~84% from an ensemble of 100 ResNet-18s (Table 2) seem very high. In comparison, ResNet-50 and AmoebaNet-A (2019) obtained a Top-1 accuracy of 77.2% and 83.9% respectively. What do the authors think about this?
- Why would one expect high accuracy of the ensemble of NNs in the acquisition model to indicate good sampling quality of the acquisition model? (caption for table 2)

Minor issues:
- Algorithm 1: first two steps should be kept un-italicized, like the rest of the steps.
- Page 7, first paragraph: "This shows that the checkpoints are obtained with no additional computational cost at train time can be used to generate diverse ensembles." The first "are" in this sentence is unnecessary.
- Build-up was chosen as the initialization scheme for the rest of the studies, as it performed best when the acquisition function was fixed at *mutual information*. However, the acquisition function that was finally chosen for the rest of the studies is *variation ratio*, since it performed best when the initialization scheme was fixed at build-up. It would be more convincing if figure 1 also includes variation ratio.

**Experience Assessment:**

I do not know much about this area.

**Review Assessment: Checking Correctness Of Derivations And Theory:**

N/A

**Review Assessment: Checking Correctness Of Experiments:**

I carefully checked the experiments.

**Review Assessment: Thoroughness In Paper Reading:**

I read the paper thoroughly.

---

> ### Author Response · Authors · 2019-11-15
> **Reply to Official Blind Review #3**
>
> We appreciate the thoughtful and helpful feedback provided by the reviewer. We address the clarifications requested in the review as follows:
>
> 1. Missing detail on the build-up initialization scheme.
>
> Please refer to Section 2.1 on Page 3 of the revised draft, where we have included additional details regarding our use of the build up initialization scheme as follows:
>
> “Finally, in the build up scheme, we follow an iterative AL loop. Specifically, we start by initializing $S$ with a randomly selected subset of the data to train an acquisition model. After performing acquisition, instead of training a single subset model, we optimize an ensemble of $E$ networks. This ensemble is used as an acquisition model for a subsequent iteration. Our goal is to finally reach a subset of $N_s$ samples. As observed by [1], exponentially growing the dataset size offers practical benefits in an AL loop setting. We therefore follow this approach, by initializing $S$ with $\frac{N_s}{8}$ random samples, and iterating two further times at $\frac{N_s}{4}$ and $\frac{N_s}{2}$ samples before obtaining a final subset of size $N_s$.”
>
> 2. Number of seeds in Tables 2 and 3.
>
> We would like to clarify that the rows ‘Single (1)’, ‘Checkpoints (5)’ and ‘Checkpoints (20)’ use the best seed of the 5 runs. We have now clarified this in the text after referencing Table 2, on Page 7.
>
> 3. High accuracy of the ResNet-18 100-model ensemble compared to state-of-the-art.
>
> We would like to highlight that the results in Table 2 are not on the official 50,000 image validation set for ImageNet, but on the 40% seen and 60% unseen data out of the 1.28M samples in the ImageNet training partition. Since the algorithm selects the harder samples to its training data, these numbers are not indicative of performance on an i.i.d validation dataset and cannot be compared directly to state-of-the-art results. Evaluating accuracy gains due to the proposed implicit ensembling technique is an interesting direction which we will look into for future research.
>
> 4. Why would one expect high accuracy of the ensemble of NNs to indicate good sampling quality?
>
> The increase in accuracy in Table 2 shows that there is a significant shift in the final prediction of the ensemble with more checkpoints, indicating diversity. Since the stored checkpoints are large in number and diverse enough to cause an improvement in accuracy, we believe they could also lead to better sampling by Monte Carlo estimation. We have rephrased the caption of Table 2 to better convey our intended point.
>
> 5. Minor typos in Algorithm 1 and Page 7.
>
> Thank you, we have made the recommended changes.
>
> 6. Comparing initialization schemes for variation ratios.
>
> Please refer to Fig. 2 on Page 13 of the revised draft. Unfortunately, we could not run the complete set of experiments from Fig. 1 using variation ratios, but we plot the validation curve for pretrain, compress and build up on 80% of ImageNet using variation ratios. We observe that the performance of all three initialization schemes is similar at 80%. The build up scheme which is used in most of our experiments in the main paper does slightly better than the compress scheme in terms of final validation accuracy.
>
> [1] Kashyap Chitta, Jose M. Alvarez, Adam Lesnikowski. Large-Scale Visual Active Learning with Deep Probabilistic Ensembles. arXiv:1811.03575, 2018.

---

### Official Review · AnonReviewer2 · 2019-10-24
**Official Blind Review #2**

**Rating:** 1

**Review:**

Review Summary
--------------
Overall, I'm not quite convinced this method would be worth the trouble to implement. On the more realistic benchmarks, they need to keep ~80% of the total dataset size and the claimed "improvement" is rather small (less than 0.6% absolute gain in accuracy, e.g. from 81.86% to 82.37% on CIFAR100 and from 72.33% to 72.78% on ImageNet). There is no runtime comparison, there are missing baselines, and most of the method development seems guided by trying out many options instead of taking a principled approach. Without these, the paper is just not ready for a top conference like ICLR.

Paper Summary
-------------
The paper considers a new take on active learning for image classification: given a large fully labeled dataset, identify a subset of the data that, when training on that subset alone, yields similar performance as training on the (much larger) full dataset. The paper focuses on "ensembles" of deep neural network classifiers as the prediction model, following Lakshminarayanan et al. (2017).

The presented method is summarized in Algorithm 1. Given a suitably initialized "acquisition model", the model makes predictions on each example in the full dataset, then ranks examples using an acquisition function to find the subset of size N_s (top N_s examples by rank) where there is most "disagreement" among the model ensemble. This subset is then used to train a "subset" model (again, an ensemble of DNNs).

Experiments consider several possible initializations, acquisition functions, and ensemble sizes. Evaluation is done using the validation sets of three prominent image classification benchmarks: CIFAR10, CIFAR100, and ImageNet (1000 classes).


Significance
------------
I don't think a successful case has been made that the proposed solution would generate significant widespread interest, because the gains demonstrated here are too minimal. Looking at the primary results in Table 2, it's really only when using 80% of the total images of imagenet or cifar100 (the most realistic benchmarks) that there is a small (<1%) absolute gain in accuracy over the simpler approach of just using the full dataset. Thus, the approach is not going to significantly reduce computational burden but adds a lot of complexity.

Novelty
-----------
The method seems new to me.


Experimental Concerns
---------------------
## E1: Need to consider runtime in evaluation

None of the figures/tables that I can see report elapsed runtimes for the different methods. To me this is the fundamental tradeoff: not how many fewer examples can I learn from, but how much faster is the method than the "standard" of using the full dataset? Showing curves of validation progress over wallclock time would be a better way to present results.

The important thing here is that even *full dataset* makes progress after each minibatch. You'd need to show progress at checkpoints for each epoch in  0.2, 0.4, 0.6, 0.8, 1, 1.2, 1.4, ....

## E2: Potential missing baseline: Random subset with balanced classes

When you select a random subset, are you ensuring class balance? If not, that seems like a more refined baseline. Perhaps won't make too much difference for cifar10, but could be important for ensuring rarer classes in ImageNet are represented well.


Presentation Concerns
---------------------

## P1: Initialization description confusing

I didn't understand the "build up" method as described in Sec. 3. How large is the subset used at each phase? How do you know when to stop? This could use a rewrite to improve clarity.

## P2: Missing some details for reproducibility

How was convergence assessed for all models? How were learning rates set? Many of these are crucial to understanding the runtime required for different models. (Sorry if these are in the appendix, but some short summary is needed in the main paper)

## P3: Title Change Recommended

I don't think the presented method is really doing a "Distribution Search"... I would suggest "Training Data Subset Selection with Ensemble Active Learning"

Minor Method Concerns
---------------------

## M1: What about regression?

Acquisition functions seem specialized to classification. What to do for regression or structure learning? Any general principles to recommend?


**Experience Assessment:**

I have read many papers in this area.

**Review Assessment: Checking Correctness Of Derivations And Theory:**

N/A

**Review Assessment: Checking Correctness Of Experiments:**

I assessed the sensibility of the experiments.

**Review Assessment: Thoroughness In Paper Reading:**

I read the paper at least twice and used my best judgement in assessing the paper.

---

> ### Author Response · Authors · 2019-11-15
> **Reply to Official Blind Review #2**
>
> Many thanks to the reviewer for their helpful suggestions. We address the main points brought up in the review as follows:
>
> 1. Need to keep ~80% of the dataset size and improvement is rather small.
>
> We would like to highlight the fact that benchmarks such as ImageNet are already highly curated by manual procedures, yet we are still able to remove 280k labeled samples that do not contribute to the performance with the proposed procedure. Previous attempts such as [1] only show that 10% can be removed, without any improvement. We demonstrate that our approach can automatically curate large datasets, with limited manual effort of a one-time implementation. Subsequently, the computation is reduced for any following training experiments that may be needed to run. We believe this is significant, since there has been a lot of emphasis lately on the financial and environmental costs of large-scale training [2,3].
>
> 2. Considering runtime in evaluation.
>
> We now show validation error curves against the number of training iterations for ResNet-18 training  on ImageNet with the full dataset vs. 80% of the data sampled by different initialization. Please refer to Fig. 2 on Page 13 of the revised draft. Our AL based sampling leads to better performance with 20% less training time than the full dataset (Full-100 in the plot).
>
> 3. Potential missing baseline: random subset with balanced classes.
>
> We thank the reviewer for suggesting this baseline. For CIFAR, we find that standard random sampling maintains class balance. On ImageNet, since some classes have only ~700 samples, we perform an evaluation of class-balanced random sampling at 10%, 20% and 40% of the data by choosing 128, 256 and 512 samples per class respectively. Our results are summarized in Table 6 on Page 14 of the revised draft. We observe no significant difference in performance between standard random sampling and random sampling with class balance.
>
> 4. Initialization description confusing.
>
> Please refer to Section 2.1 on Page 3 of the revised draft, where we have included additional details regarding our use of the build up initialization scheme as follows:
>
> “Finally, in the build up scheme, we follow an iterative AL loop. Specifically, we start by initializing $S$ with a randomly selected subset of the data to train an acquisition model. After performing acquisition, instead of training a single subset model, we optimize an ensemble of $E$ networks. This ensemble is used as an acquisition model for a subsequent iteration. Our goal is to finally reach a subset of $N_s$ samples. As observed by [4], exponentially growing the dataset size offers practical benefits in an AL loop setting. We therefore follow this approach, by initializing $S$ with $\frac{N_s}{8}$ random samples, and iterating two further times at $\frac{N_s}{4}$ and $\frac{N_s}{2}$ samples before obtaining a final subset of size $N_s$.”
>
> 5. Description of convergence and learning rates.
>
> As per the recommendation from the reviewer, we have moved the implementation details from the appendix to Section 3.1 on Page 5 of the revised draft.
>
> 6. Title change recommended.
>
> We agree with the reviewer’s comment regarding the absence of a full ‘Distribution Search’ in our approach. We have updated the title to ‘Training Data Subset Search with Ensemble Active Learning’.
>
> 7. Acquisition functions for regression.
>
> We focus on classification in our study due to several popular benchmarks being available in the large-scale setting, as well as existing work on these benchmarks from the active learning community. For Bayesian uncertainty estimation in the regression setting, acquisition functions that have been applied in the literature include variance and standard deviation [5]. We have added a note regarding this in Page 4 of the revised draft, after Eq. 5.
>
> References
>
> [1] Vighnesh Birodkar, Hossein Mobahi, Samy Bengio. Semantic Redundancies in Image-Classification Datasets: The 10% You Don't Need. arXiv:1901.11409, 2019.
>
> [2] Emma Strubell, Ananya Ganesh, Andrew McCallum. Energy and Policy Considerations for Deep Learning in NLP. Association for Computational Linguistics (ACL), 2019.
>
> [3] Roy Schwartz, Jesse Dodge, Noah A. Smith, Oren Etzioni. Green AI. arXiv:1907.10597, 2019.
>
> [4] Kashyap Chitta, Jose M. Alvarez, Adam Lesnikowski. Large-Scale Visual Active Learning with Deep Probabilistic Ensembles. arXiv:1811.03575, 2018.
>
> [5] Evgenii Tsymbalov, Maxim Panov, Alexander Shapeev. Dropout-based Active Learning for Regression. arXiv:1806.09856, 2018.

---

### Official Review · AnonReviewer4 · 2019-11-07
**Official Blind Review #4**

**Rating:** 6

**Review:**

Paper Summary:

This paper proposes a new method that uses uncertainty estimation to do ensemble active learning on image classification tasks. The proposed method mainly consists of two steps:
1. select a subset of data based on the uncertainty estimation from an ensemble model. The ranking of data point is calculated via an acquisition function, which measures the model uncertainty.
2. train a “subset model” on the selected subset.
The paper then considers 3 initialization methods, 4 acquisition functions and 4 ensemble configurations and makes an empirical study on different combinations of these three. The experiment is done progressively. The paper first fixes the acquisition function to Mutual Information and finds the build-up initialization most promising. It then fixes the build-up initialization and finds Variation Ratios and Mutual Information result in better performance. Finally, with build-up initialization and Mutual Information acquisition, the paper compares different ensemble configurations and shows checkpoints ensemble can scale up better (less computation burden) and result in better performance when the number of ensembles is large. In addition, the paper also shows the obtained subset can help models with richer capacity as well. Results on deeper architectures outperform their corresponding performance when trained on the entire training set.

Strengths:
- The idea of using checkpoints as free samples of models is new and practical.
- Experiments are comprehensive and convincing in terms of the performance.

Concerns:
- Although saving checkpoints is “cheap” and shows empirical good performance, the models are somehow dependent on each other, particularly in experiments where consecutive checkpoints are saved. I am not sure whether this fits well in the bayesian framework of uncertainty estimation.
- The build-up initialization method lacks details for reproducibility if the Algorithm 1 of Chitta et al., 2018a is used.
- It is interesting that large number of ensembles increases the accuracy of acquisition model by a lot, but doesn’t boost the performance of the subset model too much (according to table 2 and 3). Does this imply that it is the ensemble rather than the active learning that helps?

Minor issue:
- It might be better to clarify which ensemble is used in the ensemble configuration experiment since previous experiments can have two different ensembles (acquisition and subset).

**Experience Assessment:**

I do not know much about this area.

**Review Assessment: Checking Correctness Of Derivations And Theory:**

N/A

**Review Assessment: Checking Correctness Of Experiments:**

I assessed the sensibility of the experiments.

**Review Assessment: Thoroughness In Paper Reading:**

I read the paper at least twice and used my best judgement in assessing the paper.

---

> ### Author Response · Authors · 2019-11-15
> **Reply to Official Blind Review #4**
>
> We thank the reviewer for their insightful comments and suggestions. We would like to clarify the key points brought up in the review as follows:
>
> 1. Consecutive checkpoints are dependent on each other, which may not fit well with the Bayesian framework for uncertainty estimation.
>
> We agree that ideally, different samples drawn while estimating uncertainty in a Bayesian framework should be independent of each other. However, popular approaches to Bayesian uncertainty estimation exhibit high correlation between models, usually even more so than checkpoints after one epoch of training. For example, MC Dropout [1], the most widely used Bayesian approximation for deep learning, only changes the dropout mask of a single trained network to provide different samples for uncertainty estimation.
>
> 2. Build-up initialization method lacks details for reproducibility.
>
> Please refer to Section 2.1 on Page 3 of the revised draft, where we have included additional details regarding our use of the build up initialization scheme as follows:
>
> “Finally, in the build up scheme, we follow an iterative AL loop. Specifically, we start by initializing $S$ with a randomly selected subset of the data to train an acquisition model. After performing acquisition, instead of training a single subset model, we optimize an ensemble of $E$ networks. This ensemble is used as an acquisition model for a subsequent iteration. Our goal is to finally reach a subset of $N_s$ samples. As observed by [2], exponentially growing the dataset size offers practical benefits in an AL loop setting. We therefore follow this approach, by initializing $S$ with $\frac{N_s}{8}$ random samples, and iterating two further times at $\frac{N_s}{4}$ and $\frac{N_s}{2}$ samples before obtaining a final subset of size $N_s$.”
>
> 3. Ensemble rather than active learning helps since gains in acquisition model are higher than gains in subset model.
>
> We would like to highlight that in Table 2, the accuracy of the acquisition model goes up by a lot, but it also has 100x more parameters. In contrast, for Table 3, the accuracy of the subset model goes up by 1.1% with no additional parameters. This indicates that both the ensemble and active learning are useful for improving accuracy.
>
> 4. Clarification whether acquisition/subset model is being evaluated in final tables.
>
> Table 2 evaluates the acquisition model, Table 3 and Table 4 evaluate the subset model. We have updated the captions for these tables to make this clearer.
>
> References
>
> [1] Yarin Gal, Zoubin Ghahramani. Dropout as a Bayesian Approximation: Representing Model Uncertainty in Deep Learning. Proceedings of The 33rd International Conference on Machine Learning, PMLR 48:1050-1059, 2016.
>
> [2] Kashyap Chitta, Jose M. Alvarez, Adam Lesnikowski. Large-Scale Visual Active Learning with Deep Probabilistic Ensembles. arXiv:1811.03575, 2018.

---

### Decision · Program_Chairs · 2019-12-19

**Decision:**

Reject

**Comment:**

This paper proposes an ensemble-based active learning approach to select a subset of training data that yields the same or better performance. The proposed method is rather heuristic and lacks novel technical contribution that we expect for top ML conferences. No theoretical justification is provided to argue why the proposed method works. Additional studied are needed to fully convincingly demonstrate the benefit of the proposed method in terms computational cost.